# Does Abdominal Obesity Increase All-Cause, Cardiovascular Disease, and Cancer Mortality Risks in Older Adults? A 10-Year Follow-Up Analysis

**DOI:** 10.3390/nu14204315

**Published:** 2022-10-15

**Authors:** Letícia de Almeida Nogueira e Moura, Valéria Pagotto, Cristina Camargo Pereira, Cesar de Oliveira, Erika Aparecida Silveira

**Affiliations:** 1Graduate Program in Health Sciences, Medicine Faculty, Federal University of Goiás, Goiânia 74605-050, GO, Brazil; 2Graduate Program in Nursing, Nursing Faculty, Federal University of Goiás, Goiânia 74605-080, GO, Brazil; 3Department of Epidemiology and Public Health, Institute of Epidemiology and Health Care, University College London, London WC1E 6BT, UK

**Keywords:** aging, abdominal obesity, nutritional status, cardiovascular diseases, cancer, mortality

## Abstract

There is insufficient evidence on the impact of abdominal obesity (AO) on mortality in older adults. Therefore, the objective to analyze the 10-year impact of AO, assessed using different diagnostic criteria, on all-cause, cardiovascular disease (CVD), and cancer mortality in older adults. In this prospective cohort study of older adults (≥60 years), sociodemographic, lifestyle, clinical history, laboratory test, and anthropometric data were analyzed. The considered were used for AO diagnostic: waist circumference (WC) of ≥88 cm for women and ≥102 cm for men; WC of ≥77.8 cm for women and ≥98.8 cm for men; and increased waist-to-hip ratio (WHR), being the highest tertile of distribution by sex. Multivariate Cox regression and Kaplan–Meier analyses were performed. A total of 418 individuals, with an average age of 70.69 ± 7.13 years, participated in the study. In the analysis adjusted for sex and age, WHR was associated with a high risk of all-cause mortality (*p* = 0.044). Both cutoff points used for the WC were associated with an increased CVD mortality risk. None of the AO parameters were associated with cancer mortality. An increased WHR was associated to a higher all-cause mortality risk factor, while an increased WC was a risk factor for a higher CVD mortality in older adults.

## 1. Introduction

Globally, with ever-growing population ageing, non-communicable diseases (NCDs) including abdominal obesity (AO) are one of the main causes of mortality later in life [1,2]. AO is strongly related to other chronic diseases such as cardiovascular disease and cancer [3]. Several pathways contribute to the mechanism of development of these diseases, such as the release of pro-inflammatory cytokines that lead to the induction of chronic inflammation, metabolic dysfunction, cell differentiation, and angiogenesis [4].

Abdominal obesity increases progressively with advancing age [5] and its high rates across different countries and continents [6,7,8,9,10] are a great public health concern. The prevalence of AO varies between 24% in South Korea [11], between 50% and 55% in different regions of Brazil [6,7], reaches up to 68% in Norway [8], 67% in Malaysia [10], and 74% in the USA [12].

AO is characterized by the accumulation of body fat in the abdominal region [13]. Different measures or anthropometric indices, such as waist circumference (WC) and waist-to-hip ratio (WHR) can be used for its diagnosis [14]. These measures are of low cost, easy to apply, do not require any equipment, and can be used simultaneously [15]. Their use is important since body mass index (BMI) is not able to measure visceral fat [14].

Previous studies analyzing the association between AO indicators and all-cause, CVD, and cancer mortality risk produced divergent results, mainly due to differences in the criterion used for the classification of AO [16,17,18,19,20,21,22]. Some studies conducted in older adults did not find a significant association between AO, evaluated through an increased WC, and all-cause mortality [16,17,18]. Other studies conducted in older adults found a significant association between AO, defined as an increased WHR, and all-cause mortality [19,20,21]. Furthermore, different results have been observed between AO indicators and specific causes of mortality, such as CVD [23,24,25] and cancer [24,25].

There are still controversies around the causal association between AO parameters and mortality [16,17,18,19,20,21,22,23,24,25]. This could be attributed to the methodological heterogeneity observed in previous research and conflicting findings depending on the parameters used. Thus, there is a clear need for further research assessing the effect of AO and its different indicators on mortality risk in older adults [16,17,18,19,20,21,22,23,24,25]. A better understanding of this relationship could stimulate the development of public health strategies aimed at improving health care of older adults with AO. Therefore, the objective of this study was to evaluate the impact of AO assessed using different anthropometric indicators on 10-year all-cause, CVD, and cancer mortality risk in community-dwelling older adults.

## 2. Material & Methods

### 2.1. Study Population

This study used data from the Older Adults Project Goiânia (Projeto Idosos Goiânia), a prospective cohort study. Its main aim was to assess the health conditions and nutri-tional aspects of older adults resident in the city of Goiânia, Goiás, the capital of the Midwest region of Brazil. In order to ensure a representative population of older adults in the municipality, who were also users of primary health care, a sampling process was carried out. A detailed description of this process can be found in previous publications [26,27,28,29,30].

Participants in this study included 418 community dwelling older adults aged 60 and older, who carried out outpatient consultations through the Unified Health System in the last 12 months before the start of data collection. This project started in 2008; thus, the follow-up was 10.8 years. During the follow-up of this cohort, there were 25 participants (6%) who were treated as loss to follow-up and 25 (6%) who refused to participate. Those with conditions that could compromise the measurement of the anthropometric indices or those with cognitive and/or hearing impairment that could hinder their ability to answer questionnaires were excluded.

### 2.2. Study Variables

Sociodemographic, lifestyle, clinical history, anthropometric, body composition, and biochemical data were collected during home visits. In this phase, a standardized protocol of applying a structured questionnaire and obtaining the anthropometric measurements was utilized. The technical measurement error method was used to ensure a high level of inter- and intra-examiner reliability. Subsequently, a biochemical test and multifrequency electrical bioimpedance test were conducted in a nutritional assessment laboratory at the School of Nutrition of the Federal University of Goiás, Brazil.

The sociodemographic data collected were sex, age, skin color/race (white, brown, or black), marital status (living with a partner), education (schooling years), and economic class (A, B, C, D, and E) [31]. Economic class was classified using the Brazil Economic Classification Criterion (CCEB) consisting of data such as level of education and items owned by the family. Economic stratification corresponds with the economic class of the participant (A, B, C, D, and E). For statistical purposes, the classification was redefined as A/B/C, and D/E [31], with classes D/E corresponding to those from a low socioeconomic status.

The lifestyle data included alcohol consumption, physical activity, and smoking status. The question “Do you consume alcoholic beverages?” was used to assess alcohol use. Participants answered yes or no. Those who said ‘yes,’ regardless of type or quantity, were considered alcoholic beverage consumers.

The assessment of sedentary lifestyle was based on four domains: leisure-time physical activity (inactive—no leisure activity), domestic activity (inactive—no heavy domestic activity in less than three days a week lasting less than three h), physical activity at work (inactive—sitting most of the time or performing only activities with little physical effort) and physical activity while commuting (inactive—traveling by car, motorcycle, bus or less than ten min walking/biking). Sedentary participants were those who were inactive in all four domains, while non-sedentary participants were those who were active in at least one of them [32]. Never smoker, current smoker, or previous smoker were the three categories for smoking status [33].

Hypertension was defined as a systolic blood pressure (BP) of ≥140 mmHg and/or diastolic BP of ≥90 mmHg and/or use of BP-lowering drugs [34]. A triglyceride (TG) level of ≥150 mg/dL was considered an altered value; high-density lipoprotein cholesterol (HDL-c) level of <40 mg/dL for men and <50 mg/dL for women, low; and low-density lipoprotein cholesterol (LDL-c) level of ≥130 mg/dL, high [35]. Diabetes was diagnosed as a fasting blood glucose level of ≥126 mg/dL and/or HbA1c level (glycated haemoglobin) of ≥6.5% and/or use of oral blood glucose level-lowering drugs [36]. An investigation was also conducted for the presence of diabetes and hypertension as pre-existing chronic diseases. Data on the self-reported diseases were collected through the answer to the following question: “What diseases did the doctor say you have?”. As self-reported diseases, infarction, cancer, and stroke were considered.

Weight and height were measured in duplicate, and the arithmetic mean of the measurements was considered as the result. Weight was measured in kilograms using a calibrated portable digital electronic scale, with a capacity of up to 150 kg and an accuracy of 100 g. Height was measured using a 2-m measuring tape with an accuracy of 0.1 cm; the tape was fixed on a flat wall, with no baseboard but with the support of a plumb line and square [37].

The WC and hip circumference (HC) were evaluated using an inextensible measuring tape. The WC was measured at the midpoint between the superior anterior iliac crest and the last rib [37,38]. The HC was measured with the examiner positioned laterally to the participants, so that the maximum gluteal extension could be seen. The inelastic tape was passed at this level, around the hip, and in the horizontal plane, without applying pressure [38].

### 2.3. Abdominal Obesity (AO)

For AO diagnosis, three parameters were used: (i) WC of ≥88 cm for women and WC of ≥102 cm for men [38]; (ii) WC of ≥77.8 cm for women and WC of ≥98.8 cm for men, determined for the study population and published previously [28]; and (iii) an increased waist-to-hip ratio (WHR), determined by the highest tertile of the distribution according to sex.

Specifically, for WHR, although there are well-established references, such as the World Health Organization (WHO) [38], these are applicable to the general population (adults) and are not specific to the older adults. We performed statistical tests with the cited reference; however, it was not possible to continue the multivariate analysis for mortality from cancer and cardiovascular diseases due to the extremely small “*n*” value.

In previous studies that evaluated the effect of abdominal obesity on mortality in the older adults, we found the possibility of evaluating WHR by quintile [24] and in tertile [21]. Therefore, in this present study, considering the important changes in body composition that occur as a result of aging, as well as the variations attributed to ethnic differences, we chose to use tertile analysis.

### 2.4. Mortality Ascertainment

Mortality data were collected after 10.8 years of cohort follow-up. Death was confirmed during home visits through verbal autopsy by the family members, with data on the date, cause, and place of death. Subsequently, mortality records were obtained from the Mortality Information System of the Municipal Health Department (*Sistema de Informação de Mortalidade da Secretaria Municipal de Saúde*-SIM/SMS) of Goiânia-Goiás, Brazil. We did not have missing data regarding death information, since these data were obtained through the Mortality Information System national database and checked with family members (verbal autopsy).

### 2.5. Statistical Analysis

The database was structured using the SPSS software version 25.0. Data were double-entered in the same software, and all inconsistencies were subsequently checked. The STATA software version 12.0, from the company StataCorp, located in College Station, TX (**EUA**), was used for data analysis. All variables were analyzed descriptively using absolute and relative frequencies, means, and standard deviations.

The impact of AO on mortality was determined using a bivariate Cox analysis between the dependent variable (mortality) and the four parameters used to evaluate the independent variable (AO). Survival curves were plotted for the older adults with and without AO using the Kaplan–Meier method and compared statistically using the log rank test. Variables with *p*-values of <0.05 were considered statistically significant.

Subsequently, a Cox regression model was adjusted to verify the risk of AO on mortality after 10.8 years of follow-up. Variables with *p*-values of <0.20 verified in the analysis of the factors associated with obesity were adjusted in the model. We decided to use threshold of a *p*-value < 0.2 is based on the literature on traditional stopping rule and optimal *p*-values. The literature recommended a *p*-value in the range of 0.15–0.20 [39]. The results of the Cox regression analysis were presented as hazard ratios (HR’s) and their respective 95% confidence intervals (95% CI’s).

### 2.6. Ethical Aspects

The project was approved by the Research Ethics Committee of the Federal University of Goiás, according to Resolution No. 466/2012 of the National Health Council. The matrix project was approved by the Research Ethics Committee of the Federal University of Goiás under Protocol No. 031/2007 approved in 13 March 2007. The research project for the follow-up of the cohort was also approved by the Research Ethics Committee (CEP) of Hospital das Clínicas from Universidade Federal de Goiás (UFG), with protocol number No. 2.500.441/2018, approved on 19 February 2018. All participants gave informed consent, and their anonymity was preserved.

## 3. Results

### 3.1. Baseline Characteristics

A total of 418 individuals with an average age of 70.69 ± 7.13 years participated in the study. At baseline, 41% had up to four years of schooling; 47% belonged to socioeconomic class “C”; and 55% lived with a partner. Regarding lifestyle variables, 64.3% were sedentary; 84.7% did not consume alcoholic beverages; and 47.4% did not smoke. Regarding pre-existing diseases, 60.3% had hypertension, and 23.4% had diabetes. Stroke was reported by 3.6%, acute myocardial infarction by 2.2%, and cancer by 0.1% of the participants. The prevalence of AO was high, ranging from 33.2% to 75%, based on the WHR and WC, respectively. After 10.8 years of follow-up, the all-cause mortality rate was 35.2%, while the CVD and cancer mortality rates were 11.2% and 5.3%, respectively (Table 1).

### 3.2. All-Cause Mortality

In the bivariate Cox regression analysis, an increased WC was not associated with an increased risk of all-cause mortality, regardless of the cutoff point used. Only an increased WHR (third tertile) was significantly associated with an increased risk of all-cause mortality (HR: 1.62; 95% CI: 1.16–2.25) (Table 2). In the adjusted analysis, the highest WHR tertile remained significantly associated with a high risk of mortality in model 1 (HR: 1.46; 95% CI: 1.00–2.11) (Table 3).

### 3.3. CVD Mortality

In the unadjusted analysis, none of the AO indicators were significantly associated with CVD mortality. However, after adjustments, the cutoff point of WC ≥ 88 cm for women and WC ≥ 102 cm for men was significantly associated with an increased risk of CVD mortality in model 1 (HR: 2.61; 95% CI: 1.15–5.91) and in model 2 (HR: 2.45; 95% CI: 1.03–5.84). The same trend was observed with the cutoff point WC ≥ 77.8 cm for women and WC ≥ 98.8 cm for men in model 1 (HR: 2.71; 95% CI: 1.14–6.44) and in model 2 (HR: 2.88; 95% CI: 1.15–7.23). Meanwhile, the WHR did not show a significant association with CVD mortality in the adjusted analysis (Table 2).

### 3.4. Cancer Mortality

In the unadjusted analysis, none of the AO indicators were associated with cancer mortality. This result was confirmed in the multivariate analysis, in which no significant associations were found between the WC and WHR, with cancer mortality (Table 2).

### 3.5. Survival Curves

The older adults with AO based on the WHR had a shorter survival time than those without AO (*p* = 0.004) (Figure 1). No significant differences were found between the participants with and without AO based on the WC of ≥88 cm for women and WC of ≥102 cm for men (*p* = 0.461) (Figure 2), WC of ≥77.8 cm for women and WC of ≥98.8 cm for men (*p* = 0.517) (Figure 3).

## 4. Discussion

This analysis showed that an increased WHR was associated with a higher risk of all-cause mortality, while an increased WC was associated with a higher risk of CVD mortality. This study makes an important contribution to the research on the impact of abdominal obesity on mortality risk in older adults, as it used three AO indicators to assess their respective impact on mortality (all-cause, CVD and cancer) during a 10.8-year follow-up.

Our analyses showed that the highest prevalence of AO was identified when we used waist circumference (WC) i.e., 75%. The observed high prevalence of AO is similar to the results from a previous study on Asians living in the USA, in which the prevalence of AO determined using WC was 73.5% [12]. A systematic review with meta-analysis found that the prevalence of abdominal obesity was higher in studies with older individuals [40]. In older adults, the high prevalence of AO can be initially explained by the physiological changes related to aging, such as reduced basal metabolic rate [40], and the redistribution of adipose tissue and greater deposition in the visceral region [41]. However, the high prevalence of AO among our participants may reflect the increasing trend of AO later in life [42].

In this study, an increased WHR was associated with a higher all-cause mortality risk. A similar finding was found in a North American study with participants aged ≥ 60 years, in which an increased WHR corresponded to the highest risk of mortality in men and women in all BMI classification ranges [43]. The results of this study are also corroborated by a systematic review that studied data from five aging cohorts that found evidence on an increased risk of all-cause mortality in older adults with an increased WHR [19]. The relevance of using this anthropometric measure could be attributed to its ability to reflect the abundance of visceral fat over peripheral fat and muscles [44]. This assessment is necessary in older adults, considering the muscle loss and changes in fat distribution that are inherent to aging [44]. It is known that excess visceral fat is closely related to the performance of inflammatory activity and the development of obesity-related systemic diseases [45], which in this case may reflect higher mortality in those with increased WHR.

Our results agrees with the finding of a systematic review with a meta-analysis, which found that increased WC values were associated with a higher risk of mortality from CVD, regardless of BMI value [46]. However, our results differ from those of studies conducted in North American and European populations, in which an increased WC was not associated with an increased CVD mortality in older adults [24,47]. Our findings can be justified by the fact that visceral adiposity and ectopic fat contribute to adverse health outcomes such as the development of diabetes, atherosclerosis, and CVD [48]. In individuals with OA, the development of these diseases is favoured through pathophysiological mechanisms that lead to an increase in the inflammatory process, greater insulin resistance, mitochondrial dysfunction, and the secretion of adipokines that stimulate the proliferation of tumour cells [3,4].

No significant association between AO indicators and cancer mortality was found in the present study, similar to the results shown in a systematic review and meta-analysis of studies in older adults aged 65–74 years using a WC of 88 cm for women and 102 cm for men [46]. The results of this study were also similar to those of a study in individuals aged 55–69 years, in which increased WHR and WC were not associated with cancer mortality [24]. An integrative review study highlighted the association between AO and different types of cancer [3], which can occur due to chronic inflammation, which favors tumor development [49]. Although our study result is close to previous evidence, further studies are needed to further investigate the association between AO and cancer particularly because of the small number of cancer deaths in our study.

In the Kaplan–Meier analysis, an increased WHR was significantly associated with a shorter survival time than the other indicators studied. This result differs from that found in a Brazilian study conducted in participants aged ≥ 80 years, in which no significant associations were found between an increased WHR and survival time [22]. Conversely, our results are similar to that of another Brazilian study in older women aged 60 to 94 years, in which a significant association was found between a higher mortality risk and an increased WHR [50]. However, the differences between the populations studied compromise potential comparisons.

A limitation of our study was the low number of deaths from cancer, which could lead to a lack of statistical power to observe any significant association. However, the paucity of evidence on AO and cancer mortality risk made the inclusion of this information relevant [24,46], since it can contribute to knowledge in this area. Meanwhile, some strengths of this study can be highlighted, such as the extensive training of the research team in collecting anthropometric data and other methodological precautions to avoid biases. In addition, we have analyzed different AO indicators, which made our investigation different from previous research. Finally, the use of anthropometric indicators is important, as they are the most practical tools, have the lowest cost, and are available in primary care health settings.

Future research on this topic should ideally include data on history of reduction or gain in WC in adulthood or even after 65 years of age. Research in this area should be encouraged, as it can generate findings that will in turn allow the identification of older adults with AO with a higher mortality risk. This will ultimately stimulate adequate treatment aiming at reducing WC and WHR. Primary health care services are ideal settings to identify and treat AO through encouraging lifestyle changes, such as reduction of the consumption of ultra-processed foods [51], sedentary behavior and adoption of regular exercise [52], in order to avoid the development of NCDs.

## 5. Conclusions

In older adults, an increased WHR is a significant risk factor for all-cause mortality, while an increased WC is a risk factor for CVD mortality only. Owing to the particularities of aging, the use of anthropometric indicators with cutoff points specifically established for older adults may be more appropriate in assessing the risk of mortality. Older adults with increased WC and WHR should be treated in health services to reduce the prevalence of AO and consequently all-cause and CVD mortalities.

## Figures and Tables

**Figure 1 nutrients-14-04315-f001:**
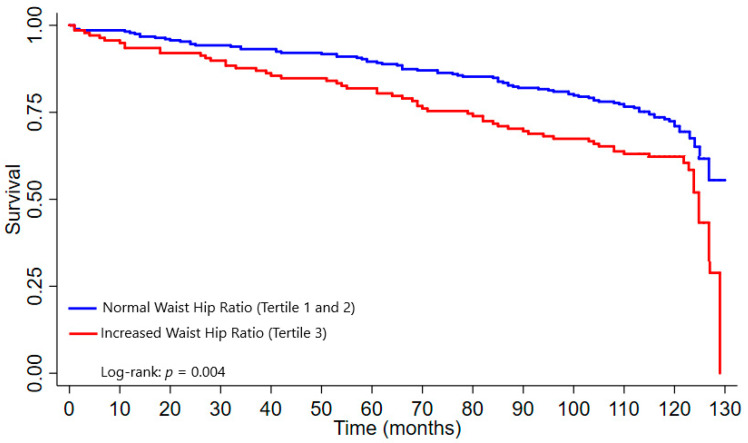
Survival curve according to Waist hip ratio (WHR).

**Figure 2 nutrients-14-04315-f002:**
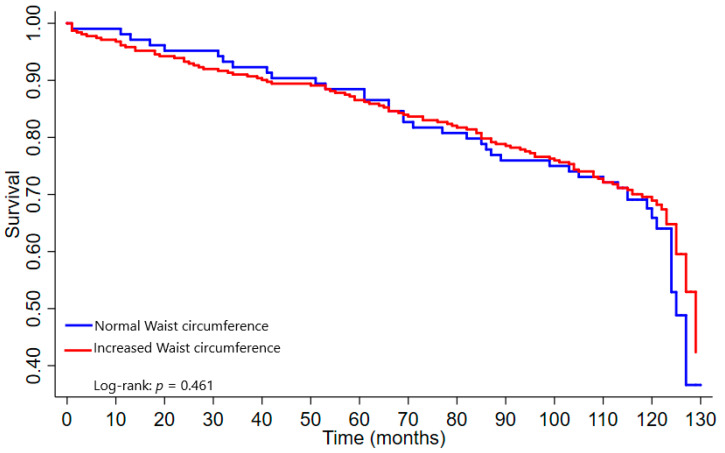
Survival curve according to waist circumference (WHO, 1995) [38]. Note: Increased waist circumference: ≥88 cm in women and ≥102 cm in men.

**Figure 3 nutrients-14-04315-f003:**
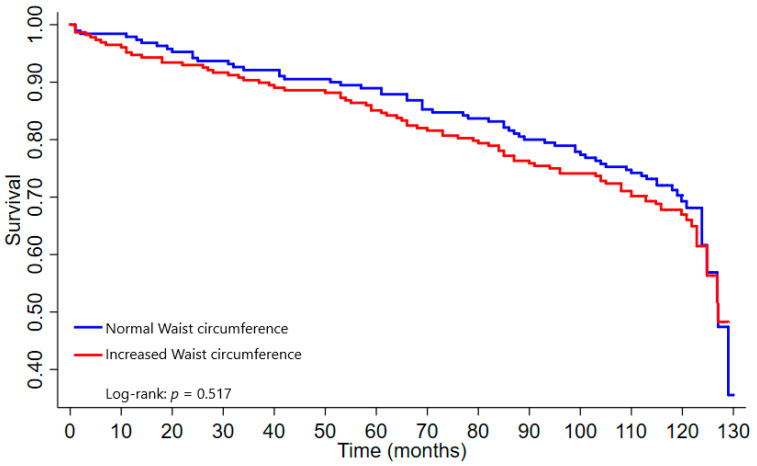
Survival curve according to waist circumference (Silveira et al., 2020) [28]. Note: Increased waist circumference ≥ 77.8 cm for women and WC ≥ 98.8 cm for men.

**Table 1 nutrients-14-04315-t001:** Distribution of participants according to sociodemographic characteristics, abdominal obesity indicators and number of deaths from all causes, CVD and cancer.

Variables	“*n*” (%)/Mean ± SD
**Gender**	
Men	142 (34.0)
Woman	276 (66.0)
**Age group (years)**	
60–69	203 (48.6)
70–79	168 (40.2)
80+	47 (11.2)
**Waist circumference (WC)**	93.49 ± 12.86
**Increased WC (WHO, 1995) [38]**	
No	104 (25.0)
Yes	312 (75.0)
**Increased WC (Silveira et al., 2020) [28]**	
No	190 (45.5)
Yes	228 (54.5)
**Hip circumference**	99.59 ± 10.30
**Waist hip ratio (WHR)**	0.94 ± 0.09
**WHR**	
1st and 2nd tertiles	278 (66.8)
3rd tertile	138 (33.2)
**Mortality-all-cause**	147 (35.2)
**Mortality-CVD**	49 (11.2)
**Mortality-cancer**	22 (5.3)

Notes: CVD: cardiovascular diseases; SD: standard deviation.

**Table 2 nutrients-14-04315-t002:** Unadjusted association between abdominal obesity indicators and causes of mortality-Cox regression.

Abdominal Obesity Indicators	Causes of Mortality					
	All-Cause		CVD		Cancer	
	HR (CI 95%)	*p*-Value	HR (CI 95%)	*p*-Value	HR (CI 95%)	*p*-Value
WC (WHO, 1995) [38]	0.83 (0.58–1.17)	0.284	1.70 (0.83–3.52)	0.149	0.85 (0.34–2.08)	0.717
WC (Silveira et al., 2020) [28]	0.87 (0.61–1.25)	0.463	2.01 (0.90–4.47)	0.089	0.91 (0.35–2.32)	0.839
WHR	1.62 (1.16–2.25)	**0.004**	1.71 (0.96–3.05)	0.067	1.57 (0.67–3.67)	0.301

Notes: CI: Confidence Interval; CVD: cardiovascular diseases; HR: Hazard Ratio; WC: waist circumference; WHR: waist hip ratio.

**Table 3 nutrients-14-04315-t003:** Multivariate Cox regression for the association between abdominal obesity indicators and of all-cause, CVD and cancer mortality risk.

Mortality Causes	WC ^a^ (Who, 1995 [38])		WC ^b^ (Silveira et al., 2020 [28])		WHR ^c^	
	HR (CI 95%)	*p*-Value	HR (CI 95%)	*p*-Value	HR (CI 95%)	*p*-Value
**All-cause**						
Model 1	0.91 (0.65–1.27)	0.567	0.92 (0.65–1.29)	0.626	1.46 (1.00–2.11)	**0.044**
Model 2	0.59 (0.35–1.02)	0.058	0.59 (0.65–1.01)	0.055	1.31 (0.77–2.23)	0.319
**CVD**						
Model 1	2.61 (1.15–5.91)	**0.022**	2.71 (1.14–6.44)	**0.023**	1.61 (0.92–2.84)	0.101
Model 2	2.45 (1.03–5.84)	**0.043**	2.88 (1.15–7.23)	**0.024**	1.34 (0.57–3.13)	0.506
**Cancer**						
Model 1	1.03 (0.37–2.42)	0.948	1.08 (0.38–3.09)	0.878	1.41 (0.56–3.55)	0.467
Model 2	1.14 (0.35–3.74)	0.830	0.90 (0.23–3.56)	0.882	2.05 (0.63–6.32)	0.230

Note: CVD: Cardiovascular diseases; HR: Hazard Ratio; CI: Confidence Interval; WHR: Waist hip ratio;.Adjusted analysis: **^a^** Model 1: sex and age; Model 2: model 1 + alcohol consumption, smoking, diabetes mellitus, stroke, adiposity, TG and HDL. **^b^** Model 1: sex and age; Model 2: model 1 + alcohol consumption, smoking, diabetes mellitus, stroke, adiposity, and TG. **^c^** Model 1: sex and age; Model 2: model 1 + alcohol consumption, diabetes mellitus, adiposity, TG and HDL.

## Data Availability

The data presented in this study are available on request from the corresponding author. The data are not publicly available to restrictions of privacy.

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
