# Peer review of "Does Abdominal Obesity Increase All-Cause, Cardiovascular Disease, and Cancer Mortality Risks in Older Adults? A 10-Year Follow-Up Analysis"

_nutrients, 2022, doi:10.3390/nu14204315_

Round 1

Reviewer 1 Report

Dear Authors,

Congratulations on your manuscript. My section-specific comments and suggestions are below.

Abstract. Please provide information on the mean and range age of the study participants.

Introduction. The information provided in this section is well known and very scarce. Nevertheless, underlying physiological mechanisms explaining the effects of AO on the increased mortality risk are not presented and not discussed in the Discussion section.

Methods, section 2.1. Please provided more detailed information on study participants' recruitment to justify the statement that the sample was representative and please specify representative to what - city, region or country? (line 63). Next, please provide the exact number of the questionnaires excluded (line 68). Were there any missing data? If yes, how they were treated?

Section 2.3. Please provide a reference justifying your cut-off choice of an increased W-H ratio as the highest tertile. Usually, in epidemiological studies, the cut-offs ≥1.0 for men and ≥0.85 for women (or similar) are used to determine an increased W-H ratio.

Section 2.4. Were any cases of loss to follow-up?

Results, section 3.1. Again, please provide information on the mean and range age of the study participants.

Table 1. Mortality (the last three rows) - probably the number and % of all-cause, CVD and cancer deaths?

Figures 1-3: please select more contrasting colours to represent differences in cumulative survival probability in WC groups.

Limitations, line 271: "...reduced number of deaths from cancer" - probably low or small number of deaths from cancer?

The reference list should meet the MDPI journals' requirements of formatting.

Author Response

Comments from the Reviewer 1

1.Abstract. Please provide information on the mean and range age of the study participants.

Response: Thank you. This information has been added in the abstract (line 21) as follows: “A total of 418 individuals, with an average age of 70.69 ± 7.13 years, participated in the study”.

2. Introduction. The information provided in this section is well known and very scarce. Nevertheless, underlying physiological mechanisms explaining the effects of AO on the increased mortality risk are not presented and not discussed in the Discussion section.

Response: Thank you. The authors have added extra information on the pathophysiological mechanisms of abdominal obesity and its effect on the aetiology of CVD and cancer in the first paragraph of the introduction (lines 32 - 35) as follows:

“OA is strongly related to other chronic diseases such as cardiovascular disease and cancer. Several pathways contribute to the mechanism of development of these diseases, such as the release of pro-inflammatory cytokines that lead to the induction of chronic inflammation, metabolic dysfunction, cell differentiation and angiogenesis” 

We have also added information on some pathophysiological mechanisms that contribute to the development of noncommunicable chronic diseases such as CVD and cancer in the fourth paragraph of the discussion (lines 285 - 289) as follows: “In individuals with OA, the development of these diseases is favoured through pathophysiological mechanisms that lead to an increase in the inflammatory process, greater insulin resistance, mitochondrial dysfunction and secretion of adipokines that stimulate the proliferation of tumour cells”

References:

  • Aparecida Silveira E, Vaseghi G, de Carvalho Santos AS, Kliemann N, Masoudkabir F, Noll M, Mohammadifard N, Sarrafzadegan N, de Oliveira C. Visceral Obesity and Its Shared Role in Cancer and Cardiovascular Disease: A Scoping Review of the Pathophysiology and Pharmacological Treatments. Int J Mol Sci. 2020 Nov 27;21(23):9042. doi: 10.3390/ijms21239042. PMID: 33261185; PMCID: PMC7730690.

  • Silveira EA, Kliemann N, Noll M,Sarrafzadegan N, de Oliveira C. Visceral obesity and incidentcancer and cardiovascular disease: An integrative review ofthe epidemiological evidence.Obesity Reviews. 2021;22:e13088.https://doi.org/10.1111/obr.13088SILVEIRAET AL.17 of 17

3. Methods, section 2.1. Please provided more detailed information on study participants' recruitment to justify the statement that the sample was representative and please specify representative to what - city, region or country? (line 63). Next, please provide the exact number of the questionnaires excluded (line 68). Were there any missing data? If yes, how they were treated?

Response: To meet the first request, we have amended the text of the first paragraph of section 2.1, specifying that the sample is representative of the city of Goiânia, capital of the state of Goiás, in Brazil, as follows (lines 65-74):

“This study used data from the Older Adults Project Goiânia (Projeto Idosos Goiânia), a prospective cohort study. Its main aim was to assess the health conditions and nutritional aspects of older adults residents in the city of Goiânia, Goiás, the capital of the Midwest region of Brazil. In order to ensure a representative population of older adults in the municipality who were also users of primary health care, a sampling process was carried out. A detailed description of this process can be found in previous publications o [26–30].”

Regarding the second point, during the follow-up of this cohort, there were 25 participants (6%) who were treated as loss to follow-up and 25 (6%) refused to participate (lines 75-76). We did not have missing data regarding death information, since these data were obtained through the Mortality Information System national database and checked with family members (verbal autopsy) (lines 155-157).

4. Section 2.3. Please provide a reference justifying your cut-off choice of an increased W-H ratio as the highest tertile. Usually, in epidemiological studies, the cut-offs ≥1.0 for men and ≥0.85 for women (or similar) are used to determine an increased W-H ratio.

Response: We understand the reviewer’s concern regarding the use of tertile distribution and adoption of the highest tertile as a reference for increased WHR. Although there are well-established references, such as the World Health Organization (WHO), which suggests WHR values > 0,90 for men and > 0,85 for women, for the assessment of increased WHR, these are applicable to the general population (adults) and are not specific to older adults. We have even performed statistical tests with the cited reference, however, it was not possible to continue the multivariate analysis for mortality from cancer and cardiovascular diseases due to the extremely small “n” value.

As it was not possible to use the references commonly used for the general population, we carried out an extensive literature review to understand how the topic is being explored in the literature. In previous studies that evaluated the effect of abdominal obesity on mortality in older adults, we found the possibility of evaluating WHR by quintile (Folsom et al., 2000) and in tertile (Bowman et al., 2017). In studies with older adults, it is necessary to use different methods, considering the important changes in body composition that occur because of aging, as well as the variations attributed to ethnic differences. Therefore, we believe that it would be more relevant to choose methods that more sensitive to the characterstics of the study population.

In the text of the manuscript, we have included a brief explanation of the option to use tertile analysis in section 2. Material & Methods, sub-item “2.3. Abdominal Obesity (AO)” (lines 138 – 147).

References:

Folsom AR, Kushi LH, Anderson KE, Mink PJ, Olson JE, Hong C-P, et al. Associations of General and Abdominal Obesity With Multiple Health Outcomes in Older Women. Archives of Internal Medicine 2000;160:2117. https://doi.org/10.1001/archinte.160.14.2117

Bowman K, Atkins JL, Delgado J, Kos K, Kuchel GA, Ble A, et al. Central adiposity and the overweight risk paradox in aging: follow-up of 130,473 UK Biobank participants. The American Journal of Clinical Nutrition 2017;106:130–5. https://doi.org/10.3945/ajcn.116.147157.

5. Section 2.4. Weny cases of loss to follow-up?

Response:  This information has been added to item 2 of the revised Material and Methods, sub-item 2.1. Study population (lines 75-76): “During the follow-up of this cohort, there were 25 participants who were treated as loss to follow-up” in the results text.

6. Results, section 3.1. Again, please provide information on the mean and range age of the study participants.

Response:  Thank you. The requested information has been added as “with an average age of 70.69 ± 7.13 years” (line 183).

7. Table 1. Mortality (the last three rows) - probably the number and % of all-cause, CVD and cancer deaths?

Response: Thank you. The first Table of the manuscript brings in its last three lines the number of deaths from all causes, cardiovascular diseases and cancer. In order to make it clearer, we have changed the title of this Table in the manuscript to: “Table 1.  Distribution of participants according to sociodemographic characteristics, abdominal obesity indicators and number of deaths from all causes, CVD and cancer” (lines 192-193). In the first line of the Table, we have also clarified that the values are described in total value (n) and percentage (%), or mean and standard deviation

8. Figures 1-3: please select more contrasting colours to represent differences in cumulative survival probability in WC groups.

Response: Thank you for the suggestion. We have changed the colors green and pink to blue and red, to have greater contrast and improve the figures’ display (Lines 238-248).

9. Limitations, line 271: "...reduced number of deaths from cancer" - probably low or small number of deaths from cancer?

Response: Thank you. We have replaced the word “reduced” with “low” (line 308).

10. The reference list should meet the MDPI journals' requirements of formatting.

Response: We carefully review the formatting of the references to fulfil the Nutrients’ requirements.

Reviewer 2 Report

Abdominal obesity and mortality is an important research topic, especially in the aging population.

My two major concerns of the present study are 1) the sample size, and 2) study not stratified by sex given the imbalance of male and female participants. The authors may consider, at least, to present the cause of death by age and sex in the appendix to show if the conclusion is likely to be biased by that.

Two minor concerns:

1. the conclusion on all-cause mortality might be over-stated. In the abstract, it stated that "in the fully adjusted analysis....". I might get it wrong, however, from my reading of the results section, that p value is actually from model 1 of the WHR analysis, which adjusts for age and sex only. It is understandable that it is hard to yield statistical significant given the sample size, however, it might be good to stick to the same standard reporting for different abdominal obesity definition and cause of death. On contrary, the non-significant findings for cancer mortality might simply due to small sample size.

2. For results in Table 3. To make a fair comparison between different abdominal obesity definitions, it might be a good idea to control for the same sets of variables.

Author Response

Comments from the Reviewer 2

  1. the conclusion on all-cause mortality might be over-stated. In the abstract, it stated that "in the fully adjusted analysis....". I might get it wrong, however, from my reading of the results section, that p value is actually from model 1 of the WHR analysis, which adjusts for age and sex only. It is understandable that it is hard to yield statistical significant given the sample size, however, it might be good to stick to the same standard reporting for different abdominal obesity definition and cause of death. On contrary, the non-significant findings for cancer mortality might simply due to small sample size.

Response: Thank you. We have made the necessary adjustments to the text of the abstract, indicating that the analysis was adjusted only for sex and age (Line 22).

  1. For results in Table 3. To make a fair comparison between different abdominal obesity definitions, it might be a good idea to control for the same sets of variables.

Response: In Table 3 of this manuscript, the analyses were stratified by the three diagnostic criteria for abdominal obesity used in the present study. For model 1, the same variables were used, which were sex and age. However, for model 2, the same variables were not used, because we have used the p-value as a criterion for modelling. Thus, those variables that in the bivariate analysis had a p value lower than 0.20 were included in the multivariate analysis model. Therefore, for model 2, of all OA parameters, the variables are different.

We decided to use threshold of a p-value < 0.2 is based on the literature on traditional stopping rule and optimal p-values. The literature recommend a p-value in the range of 0.15–0.20 (Hosmer & Lemeshow, 2013). We have included the rationale for adopting a value < 0.2 in the text of section 2. Material &Methods, sub-item 2.5. Statistical analysis (lines 170 – 172).

Reference:

Hosmer DW, Lemeshow S, Sturdivant RX. Applied logistic regression. New York: John Wiley & Sons, Incorporated, 2013.

Round 2

Reviewer 1 Report

Dear Authors,

Thank you for taking into account my comments. I appreciate that.